# Improving Bone Health in Patients with Metastatic Prostate Cancer with the Use of Algorithm-Based Clinical Practice Tool

**DOI:** 10.3390/geriatrics7060133

**Published:** 2022-11-24

**Authors:** Kamal Kant Sahu, Eric D. Johnson, Katerina Butler, Haoran Li, Kenneth M. Boucher, Sumati Gupta

**Affiliations:** 1George E. Wahlen Department of Veterans Affairs Medical Center, Salt Lake City, UT 84148, USA; 2Huntsman Cancer Institute, University of Utah, Salt Lake City, UT 84101, USA; 3Intermountain Health Care, Salt Lake City, UT 84102, USA

**Keywords:** bone health, prostate cancer, bone modifying agents, bisphosphonates, zoledronic acid, denosumab

## Abstract

Background: The bone health of patients with locally advanced and metastatic prostate cancer is at risk from treatment-related bone density loss and skeletal-related events from metastatic disease in bones. Evidence-based guidelines recommend using denosumab or zoledronic acid at bone metastasis-indicated dosages in the setting of castration-resistant prostate cancer with bone metastases and at the osteoporosis-indicated dosages in the hormone-sensitive setting in patients with a significant risk of fragility fracture. For the concerns of jaw osteonecrosis, a dental evaluation is recommended before starting bone-modifying agents. The literature review suggests a limited evidence-based practice for bone health with prostate cancer in the real world. Both under-treatment and inappropriate dosing of bone remodeling therapies place additional risks to bone health. An incomplete dental work up before starting bone-modifying agents increases the risk of jaw osteonecrosis. Methods: We created an algorithm-based clinical practice tool to minimize the deviation from evidence-based guidelines at our center and provide appropriate bone health care to our patients by ensuring indication-appropriate dosing and dental screening rates. This order set was incorporated into the electronic medical record system for ordering a bone remodeling agent for prostate cancer. The tool prompts the clinicians to follow the appropriate algorithm in a stepwise manner to ensure a pretreatment dental evaluation and use of the correct dosage of drugs. Results: We analyzed the data from Sept 2019 to April 2022 following the incorporation of this tool. 0/35 (0%) patients were placed on inappropriate bone modifying agent dosing, and dental health was addressed in every patient before initiating treatment. We compared the change in the practice of prescribing and noted a significant difference in the clinician’s practice while prescribing denosumab/zoledronic acid before and after implementation of this tool [incorrect dosing: 24/41 vs. 0/35 (*p* < 0.00001)]; and an improvement in pretreatment dental checkup before and after implementation of the tool was noted to be [missed dental evaluation:12/41 vs. 0/35 (*p* < 0.00001)]. Conclusion: We found that incorporating an evidence-based algorithm in the order set while prescribing bone remodeling agents significantly improved our institutional clinical practice of indication-appropriate dosing and dental screening rates, and facilitated high-quality, evidence-based care to our patients with prostate cancer.

## 1. Introduction

Prostate cancer (PCa) is the most common malignancy diagnosed in Veterans in the United States [1]. Most cases of PCa are detected as a localized disease (73%), followed by regional spread (14%) and distant spread (7%). The recent report from the Surveillance, Epidemiology and End Results (SEER) database shows that the incidence rates of metastatic prostate cancer have increased in the last decade in men across all age groups [2]. The study on the age distribution pattern suggests that 60% of the patients diagnosed with new prostate cancer are 65 years and above. For localized and regional PCa, the 5-year survival is >99% [3]. Once metastatic, the 5-year survival declines significantly to approximately 30%. However, with the new treatment options, the median overall survival (OS) of patients with mPCa is also on the improving trend [4]. Currently, the median age of patients with PCa at death is 80 years.

Hence, the rising trend in the incidence rates of new cases and improved survival has led to a significant rise in the patients of PCa among the male survivorship population in the United States [5,6]. Currently, patients with PCa represent approximately 45% of the seven million survivors [7]. The American Cancer Society (ACS) and the American Society of Clinical Oncology (ASCO) developed the prostate cancer survivorship care guidelines in the year 2014 and 2015, respectively [8,9]. Hence, the last decade has seen greater attention from policymakers, cancer care advocates, and various national cancer societies regarding prostate cancer survivorship [10].

Bone health is one of the essential components of adult cancer survivorship [11]. Aging, and use of androgen deprivation therapy (ADT) are among the many factors that may lead to osteopenia and osteoporosis in patients with PCa [12]. The non-cancer-related risk factors like insufficient calcium intake, vitamin D deficiency, smoking, and alcohol abuse also add up to the development of osteoporosis and fractures. Because most patients will not ultimately die from PCa, but rather from other medical issues, treatment-related adverse events must be considered and prevented [13,14].

Moreover, patients with PCa are at a high risk of bone-related complications such as pathological fracture, painful bone lesions requiring surgery or radiation, or spinal cord compression, collectively referred to as skeletal-related events [SREs] [15]. SREs are sentinel events that cause pain, impair quality of life, function, and survival, and increase healthcare utilization [16,17].

## 2. Need of the Study: Barriers to Adhering to Guideline-Based Implementation of Bone Modifying Agents

The ASCO endorsed the Cancer Care Ontario (CCO) guidelines on Bone Health and Bone-Targeted Therapies for Prostate Cancer in 2020 [18]. The guidelines aim to guide oncologists toward the optimal use of bone-modifying agents (BMAs) in men with PCa in various settings based on evidence gathered from several studies conducted to guide indications for therapy with BMAs [19,20]. Despite strong evidence, the implementation of these guidelines remains inconsistent [21,22]. There has been frequent overuse of BMAs at the time of diagnosis without the setting of osteopenia/osteoporosis, at the time of initiation of ADT, or in treatment naïve individuals with bone metastasis [23].

Another challenge while prescribing BMAs is the potential side effect of osteonecrosis of the jaw (ONJ) [24]. The incidence of ONJ is approximately 1% to 9% of patients with advanced malignancies [25]. While most of the existing literature has focused on the continuing concerns of the appropriate use of BMAs, there is incomplete understanding or underdeveloped research on ONJ. Alleviating the risk for ONJ through screening dental care reduces the BMA treatment interruptions and improves the quality of life while using these therapies [26]. The Multinational Association of Supportive Care in Cancer/International Society of Oral Oncology (MASCC/ISOO) and ASCO have established clinical practice guideline for the prevention and management of medication-related ONJ [25]. Dental screening is a strong recommendation for oncologists to request before prescribing BMAs. However, there is a significant proportion of oncologists who still do not refer patients for a dental screening [27,28]. Hence, there are multiple factors at various steps that can lead to inappropriate use of BMAs (Figure 1).

Acknowledging these inadequacies, we identified an urgent need for further evidence-based research to improve the value of comprehensive cancer care, including bone health in our patients with PCa at the George E. Wahlen Veterans Affairs Medical Centre (VAMC) in Salt Lake City, Utah. This quality improvement project aimed to improve care excellence, provide evidence-based education to clinicians, and reduce unnecessary medical costs. Inadequate screening for osteoporosis/osteopenia at the time of ADT initiation has been recognized as a practice gap [29] which we found did not need to be addressed in our practice due to excellent rates of screening for osteoporosis/osteopenia at the time of initiation of ADT. We believe that this high rate of compliance is due to a prompt within our electronic order set to consider bone modifying agents while ordering ADT.

## 3. Materials and Methods

### 3.1. Developing a Quality Improvement Tool

At our VAMC, we established a panel of health professionals, including oncologists, pharmacists, nurse practitioners, dentists, and registered nurses. The panel discussed the evidence-based expert consensus recommendations from multiple sources (ASCO, MASCC, ISOO, and CCO) regarding bone health in prostate cancer. Recommendations from these guidelines were included in a quality improvement model, and an algorithm-based clinical practice tool was developed.

### 3.2. Stepwise Details about the Intervention

Bone health is one of the essential aspects of the treatment plan for men with PCa. Below is the detailed stepwise explanation of the various steps which are followed at our institute:

Step 1: The first step for any patient with PCa with a plan to start ADT is to document if there are bone metastasis(es) or not, which is done by bone scan conventionally.

Step 2: All patients with bone metastasis(es) will be further investigated for disease status (castration-sensitive versus -resistant). For metastatic castration-resistant prostate cancer (mCRPC), all patients get BMAs after dental screening.

Step 3: Patients with metastatic castration- sensitive prostate cancer (mCSPC), undergo a dual energy X-ray absorptiometry (DEXA) scan to assess bone density, stratifying men with normal bone health, and those with osteopenia/osteoporosis.

Step 4: Patients with osteopenia/osteoporosis are subsequently assessed for their fracture risk as per their Fracture Risk Assessment Tool (FRAX score). For a high FRAX score, patients are set up for a dental clearance followed by prescription for BMAs.

Step 5: Patients with normal bone density or low FRAX score are re-assessed for fracture risk every 2 years.

The above-mentioned steps are followed in an algorithmic tool to ensure that patients receive BMAs as per guidelines (Figure 2). As per the algorithm, all patients were stratified into one of the three order sets: (1) New osteopenia/osteoporosis (Appendix A). (2) Renew osteopenia/osteoporosis (Appendix A). (3) Bone metastasis at baseline (Appendix A). All the patients were prescribed BMA and underwent dental screening evaluation as indicated per the standard guidelines.

While prescribing the BMAs, the tool included a multistep process including a few hard stops (1) Algorithm prompt mandating to get a comprehensive oral evaluation and comprehensive dental screening before commencing BMA therapy to reduce the risk of developing ONJ. (2) Hard stop with inability to prescribe BMAs to men with mCSPC without a DEXA scan documenting a high FRAX score.

### 3.3. Study Design and Patients’ Selection

#### 3.3.1. Pre-Implementation Phase (July 2017–August 2019)

To understand the current practice at our institute, we performed a retrospective chart review, and included all the men with PCa who were followed in our oncology clinic from July 2017 till August 2019. We captured the data on the indications for the use of BMAs, and if dental health was addressed before initiation of bisphosphonate or denosumab. The aim was to utilize this data to compare with practice patterns following the implementation of the quality improvement tool and assess for significant changes.

#### 3.3.2. Implementation of Quality Improvement Tool (September 2019)

The algorithmic tool was activated in September 2019 within the electronic medical record (EMR) as an order set. This tool was intended to be utilized while prescribing a bone remodeling agent in the setting of PCa. The tool supported the BMA treatment with appropriate dosing as per the indication and prompted the pretreatment dental screening evaluation. All the oncology practitioners at our center were informed about this tool and educated about the steps to follow to prescribe a BMA. A point of contact (KB) from the expert panel was appointed for any further clarification or to assist in navigating through the steps of the algorithm in case needed.

#### 3.3.3. Post-Implementation Phase (September 2019–April 2022)

We prospectively evaluated the newly established patients with PCa who were treated with BMAs. We also evaluated the EMR for dental clearance before the prescription of BMAs.

## 4. Review of the Orders for the Appropriateness

All the orders were reviewed before releasing the prescriptions by our pharmacist (KB) who was involved in the tool development as well. The prescriptions/orders were evaluated for, (1) appropriate use: whether the indication was correct or not, and (2) appropriate dose: if correctly indicated, the dose of BMA was correct or not?

### 4.1. Indications of BMA Use

Men with mCSPC with bone metastasis(es) and DEXA scan showing osteopenia/osteoporosis with a high FRAX score (≥20% for all osteoporotic fractures or ≥3% for hip fractures).

Men with mCSPC or mCRPC with skeletal-related events (SREs) defined as defined as any of the following: pathologic fracture, radiation therapy to bone, surgery to bone, or spinal cord compression.

### 4.2. Dose of BMA Use

At our center, we used zoledronic acid or denosumab as BMA during this study period. Recommended doses were:

#### 4.2.1. Osteoporosis/Osteopenia

-Zometa 4 mg once every year-Denosumab 60 mg every 6 months

#### 4.2.2. Bone Metastasis and/or SREs

-Zometa 4 mg every 3 months.-Denosumab 120 mg monthly

## 5. Statistical Analysis

We used descriptive statistics to describe the demographics, disease, and treatment characteristics of the patients studied in the prospective phase of the study. The difference in the appropriateness of BMA prescription between pre- and post-tool implementation was compared using Fisher’s exact test. Similarly, referral to the dental clinic for screening before starting BMA was also compared between the two groups.

## 6. Results

### 6.1. Pre-Implementation Phase (July 2017–August 2019)

In our pre-implementation data gathering of 41 patients, we noted that 24 patients (58.5%) were given the correct dose of BMAs, while the rest, 17 patients (41.5%) received BMAs in doses not supported by evidence-based guidelines. Similarly, a dental assessment was rightly performed in 29 patients but not in the remaining 12 patients.

### 6.2. Post-Implementation Phase (September 2019–April 2022)

In the post-implementation period, 35 patients were enrolled till the last day of study evaluation (April 2022). As per the algorithm, all patients were stratified into one of the three order sets: (1) New osteopenia/osteoporosis (Appendix A). (2) Renew osteopenia/osteoporosis (Appendix A). (3) Bone metastasis at baseline (Appendix A). All the patients were prescribed BMAs and underwent dental screening evaluation as indicated per the standard guidelines.

### 6.3. Comparative Analysis (Outcomes Measured)

We compared the change in the practice of prescribing BMAs before versus after the execution of this tool. We noted a significant reduction (by 58.53%) in inappropriate dosing while prescribing BMAs after the implementation of this tool [24/41 vs. 0/35 (*p* < 0.00001)] (Figure 3A). We also noted a significant improvement (by 29.26%) in a lapse in dental health awareness after implementation of the tool [12/41 vs. 0/35 (*p* < 0.00001)] (Figure 3B). The Fisher’s Exact Test was used for both comparisons.

## 7. Discussion

In this quality improvement project conducted in the George E. Wahlen VAMC in Salt Lake City, Utah, we implemented an algorithmic tool for the appropriate use of BMAs in patients with prostate cancer based on the well-established guidelines of various oncology societies. Our QI project was highly successful in addressing the gap in guidance-based clinical practice of appropriate BMA use and dental precautions observed prior to implementation of the QI tool. The project was conceptualized from the observation that the quality of bone health care is suboptimal in many patients receiving cancer care for prostate cancer [21,23,30]. To reduce potential side effects of bone modifying agents, and improve quality of care, we pursued a review of the use of bone modifying agents in veterans treated for PCa at our institution. We identified an area of improvement that primarily focused on applying evidence-based use of bone modifying therapy at the appropriate dose for the correct indication, and mitigation of the risk of ONJ. An algorithm was created within the medical record ordering system as an order set with prompts predicated on best evidence-based practice. The order set guides clinicians to aid decision-making and verification of indications and clearance for use. By implementing an evidence-based algorithm and clinical practice tool, while prescribing bone remodeling agents to patients with PCa, we significantly improved our institutional practice to a high-quality, evidenced-based approach when addressing prostate cancer bone health care.

The treatment paradigm for prostate cancer is rapidly evolving, and we are seeing trends of significant improvement in the OS in patients with advanced prostate cancer [31,32,33]. Overall, this improvement in OS is encouraging, but at the same time, this also brings more accountability as more patients are now in the survivorship pool than before [34]. As per the SEER recent data, 2022, prostate cancer is the most prevalent cancer (n = 3,523,230) amongst males in the United States [34].

Patients with prostate cancer need a more all-inclusive approach, addressing various domains of well-being, other than cancer treatment. Ensuring optimal bone health during and beyond the cancer treatment is crucial to prevent the chronic morbidity and mortality related to osteopenia/osteoporosis and SREs [35,36]. Francini et al. recently studied the association of concomitant BMAs with OS and time to first SRE among patients with mCRPC and bone metastases receiving abiraterone acetate with prednisone as first-line therapy [37]. Addition of BMA was associated with significantly longer OS (31.8 vs. 23.0 months; hazard ratio [HR], 0.65; 95% CI, 0.54–0.79; *p* < 0.001). This OS benefit was found to be greater for patients with high-volume vs. low-volume disease (33.6 vs. 19.7 months; HR, 0.51; 95% CI, 0.38–0.68; *p* < 0.001) [37]. Similarly, a post hoc analysis of study COU-AA-302 was done by Saad et al. to study the role of BMAs in chemotherapy naïve mCRPC treated with abiraterone. The study showed clinical benefits of abiraterone increased with concomitant BMA with improved OS (HR 0.75; *p* = 0.01), increased time to performance status deterioration (HR 0.75; *p* < 0.001) and time to opiate use for cancer-related pain (HR 0.80; *p* = 0.036) when compared without BMA use [38]. In contrast, there are other studies which did not find clinical benefits (OS or pain response or prevention of SREs) in mCSPC or mCRPC setting with the BMA use [39,40,41,42]. These contradictory observations suggest inconsistent results with regard to the benefits (especially OS benefit) with BMA use. Hence, clinical use of BMA should be weighed against the increased risk of renal impairment, ONJ and gastrointestinal upsets in patients receiving bisphosphonates.

In real-world clinical practice, there is cumulative evidence that highlights the need to improve bone health care in patients with prostate cancer [21,23,27,28]. Mitchell et al. did a retrospective study utilizing the SEER database, including the patients with mCSPC, to evaluate the BMA prescribing patterns amongst oncologists and urologists [23]. Overall, 18.4% and 23.6% received a BMA within 90 and 180 days of diagnosis, respectively. The trend of utilization of BMA was noted to have increased from 17.3% in the 2007–2009 time period to 28.1% in the 2012–2015 time period. An overuse of BMA was noted in more than one-quarter of patients with mCSPC, even with no evidence of high osteoporotic fracture risk. Mitchell et al. also did a similar study in the mCRPC cohort to investigate the BMA utilization pattern within 180 days of starting cancer treatment [21]. 705/1034 [68%] of the patients with mCRPC and bone metastasis were given a BMA within 180 days of initiating cancer treatment. Among mCRPC patients without bone metastasis, only 26% with high fracture risk were given a BMA. Both these studies were conducted utilizing the SEER database and represent the general pattern of oncology practice across the whole United States. These two studies showed that both under and over utilization of BMA is a matter of concern and needs attention at both ends. While underutilization of BMAs can lead to serious SREs, overutilization has the potential for unnecessary adverse events (hypocalcemia, ONJ, atypical fractures, etc.) and financial toxicities [43,44,45,46]. Pre-implementation chart review in our study also showed inappropriate usage of BMAs in 41.5% of cases. With the tool’s implementation, we could recommend guidelines-based prescriptions of BMAs in 100% of our patients.

Assessment of bone health should be done in all patients with PCa at the initiation of ADT. However, in clinical practice, many patients are not offered DEXA scan for screening of osteopenia/osteoporosis. This pattern of non-compliance to bone health screening with DEXA scan is worrisome for the fact that there are effective anti-resorption therapies available that can delay bone resorption. A recent report from a SEER and Medicare data, showed that <15% of patients on ADT are being screened for osteopenia/osteoporosis. DEXA scan is a minimally invasive, readily available at most of the VA centers, cost effective test. Hence, we believe that there should be a conscious effort amongst the oncologists to ensure that DEXA scan is ordered before starting ADT therapy. In our algorithmic protocol, we included DEXA scan as well, that prompted the oncologists to follow the guidelines and assess the bone health before prescribing ADT. Our QI project was focused upon appropriate and safe use of BMAs in prostate cancer patients. There exists a gap in use of DEXA scans in patients initiating ADT for prostate cancer. We assessed our practice with regard to this aspect and did not find this gap as all patients who initiated ADT were assessed for their bone density as clinically indicated. Our system prompts the need for assessment for BMAs at the time of ADT initiation

Excellent collaboration between oncologists and dentists is needed to ensure that all patients who qualify to receive BMAs also get a preventive dental screening [47]. However, in the real world, there is a substantial evidence-practice gap among the physicians [5]. Yamori et al., in their questionnaire survey, noted that only 30% of physicians requested dental screening before initiating BMAs [27]. Similarly, Taguchi et al. reported that 62% of physicians did not seek dental clearance before prescribing BMAs [28]. In our study, we also noted that before the tool implementation, 29.27% of the patients were not sent for dental checkups before prescribing BMAs. A few of the potential causes which may likely be responsible in non-referral to dentists are (1) lack of knowledge about the guidelines (2) Non-availability of a dentist in the VA system requiring referral to the out of network community hospital (3) delay in getting timely appointment. In our institute, we held a staff meeting to ensure we appropriately addressed these issues. SG (primary investigator) presented the project details, including the guidelines for proper use of BMA to the other practicing oncologists, pharmacists, and nurses. This ensured that all the medical personnel were made aware of the quality improvement project for proper implementation of the tool. At our institute, we also have a comprehensive dental care clinic that provides timely appointments to our patients and dental assessment at the earliest available date. Another reason for the success of our study was close involvement of our pharmacist (KB) who followed all the patients with PCa individually and scanned through the charts to ensure quality care.

Overall, we believe that this algorithm tool is easy to implement at any institute. The presence of a comprehensive team (oncologists, dentists, pharmacists, nurses, etc.) is essential for the success of this project. Absence of in-house dental care could pose a potential challenge to get a timely appointment for dental screening. Additionally, regularly assessing the impact of process improvement could address any immediate concerns and may allow a timely amendment during the study period. At the end, every institute has its own unique challenges that may require specific modification in the system (Figure 1).

## 8. Limitation of the Study

There are several limitations to the present study. The study has a relatively small number of patients in each arm. However, it was sufficient to analyze the significance of the difference in the change in clinical practice. Second, the study was focused on patients with prostate cancer. However, the study’s findings suggest that the tool can be modified and applied in other advanced cancers as well.

## 9. Conclusions

In combination with the current literature, our study findings support the need to explore the practice patterns of oncology practitioners in various clinical settings regarding bone health. The poor compliance with guideline endorsed bone-health care risks the overall survival of patients with PCa. In addition, it could lead to fatal SREs and adds to avoidable financial toxicities. Simple interventions like incorporating an algorithmic tool like ours could improve the likelihood of utilizing evidence-based guidelines in actual practice.

## Figures and Tables

**Figure 1 geriatrics-07-00133-f001:**
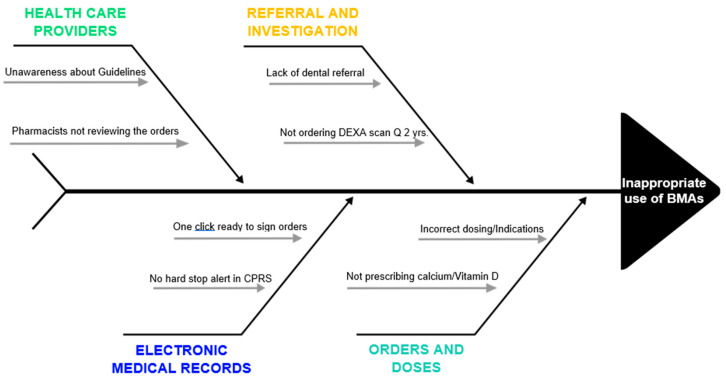
Fishbone diagram depicting pitfalls related to providers, electronic medical records, referral, investigation, and orders that contribute to inappropriate BMA use.

**Figure 2 geriatrics-07-00133-f002:**
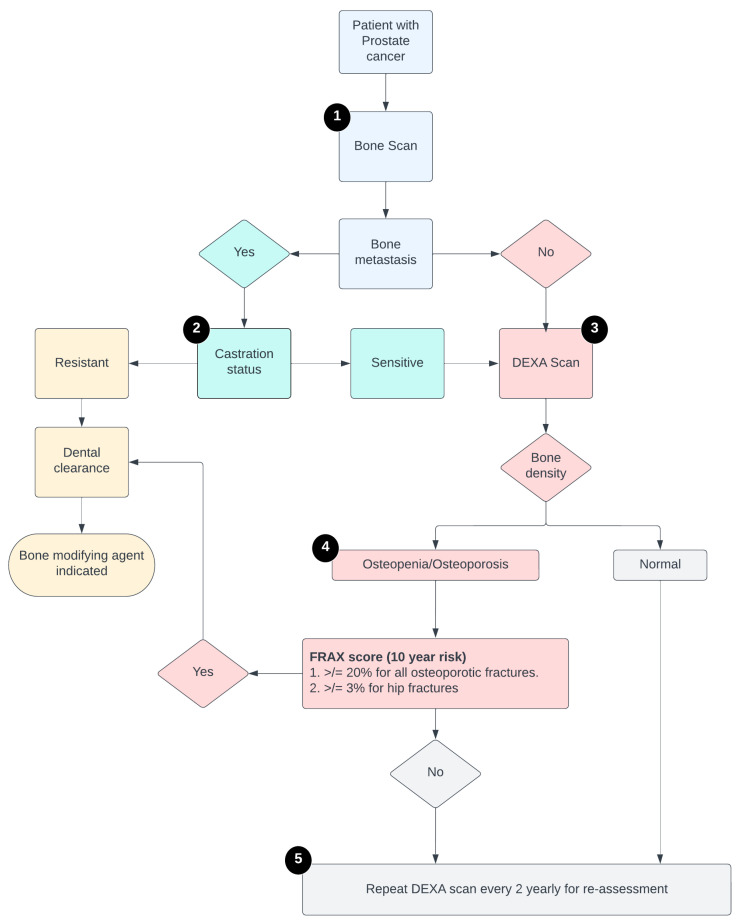
Schema of the steps of the algorithmic tool to follow before prescribing ADT.

**Figure 3 geriatrics-07-00133-f003:**
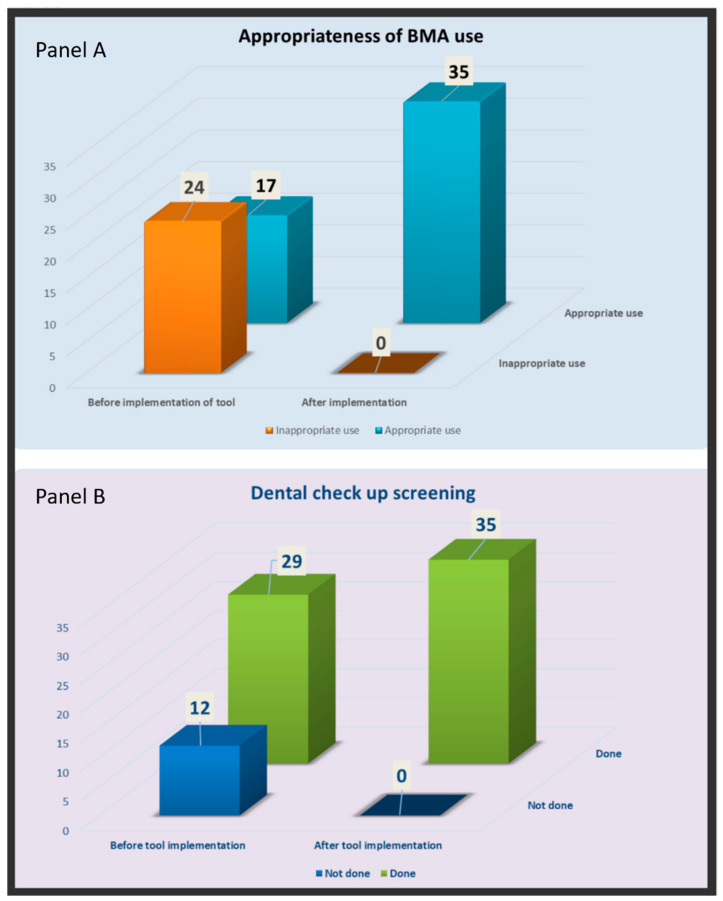
Comparing the change in the practice while prescribing BMA (**panel A**), and referral for dental screening (**panel B**) with the implementation of new algorithmic tool into Computerized Patient Record System.

## Data Availability

Data supporting reported results can be provided upon request.

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
