# Peer review of "Improving Bone Health in Patients with Metastatic Prostate Cancer with the Use of Algorithm-Based Clinical Practice Tool"

_geriatrics, 2022, doi:10.3390/geriatrics7060133_

Round 1
Reviewer 1 Report
This is a quality improvement study of a practice intervention to increase the guideline-concordant us of bone modifying agents in prostate cancer. My sense is that the underlying work done in this project is important, useful, and likely worthy of publication. However, before publication the manuscript can be substantially improved by reorganization and revision to improve clarity.
Major comments:
1. The Methods section should be substantially expanded. In particular, the algorithmic tool itself belongs in the Methods rather than Results. And much more detail is needed regarding how the tool identified appropriate patients, and what kinds of guidance (including any “hard stops”) that it provided to clinicians.
2. The Methods section (particularly “post implementation phase”) needs to be substantially more specific in terms of what is being measured. “We prospectively followed the newly established patients” is about as general as you can get. What defined inappropriate vs. appropriate use? Or an appropriate dose? Whether dental screening was done should be clarified as one of the primary study outcomes. This could be substantially clarified by creating a new subsection of “outcomes measured,” rather than repeating what was measured in both the pre-implementation and post-implementation subsections.
3. How a “correct dose” of a BMA was defined needs much more clarity. My best guess is that these were patients who did not warrant the higher SRE doses because they were still castration sensitive, but did warrant the lower doses indicated for osteoporosis. But the authors should not leave the reader guessing. Whether oral bisphosphonates were included should be clarified. In particular, how dosing of zoledronic acid was defined as inappropriate vs. appropriate should be clarified, because for this drug SRE vs. osteoporosis dosing differ not by milligrams per dose, but by the time interval between doses.
4. The figures that show the algorithm in terms of screen shots are of limited usefulness. These would be appropriate for supplementary materials, but not as main-body figures. They leave too much work left for the reader to figure out what is going on and how the clinicians are being guided. Instead, the algorithm should be described in more detail in the Methods section.
5. A substantial limitation of the study is that it only assessed patients for whom BMAs were ordered. It therefore cannot determine the proportion of patients who WOULD HAVE benefited from a BMA but never had it ordered. A particular concern I have is that be introducing additional effort for the patient by requiring a dental evaluation in all cases, this may reduce the number of patients who agreed to BMA therapy. In line with this, I note that approximately 1.6 patients per month started BMAs in the pre-intervention time period, but only about 1.1 per month in the post-intervention period. This may be random chance, but it does raise the concern about whether relatively fewer patients with prostate cancer were started on BMAs due to the increase clinician and/or patient effort. This would have been a helpful “balancing measure” to assess.
6. The study spends substantial time in the Intro and Discussion talking about overuse and underuse of BMAs – ie, the proportions of patients who received BMAs out of a given denominator of prostate cancer patients. But because, as noted above, this study did not measure the proportions of patients receiving BMAs, this focus on overuse and underuse is a bit misplaced/irrelevant to the current study. The intro and discussion should focus more narrowly on the things that this study DID address – mainly, indication-appropriate dosing and dental screening rates.
7. The Discussion is largely a repetition of the introduction. All of that which is repetitive can be cut out, and/or moved to the intro. Instead the authors should spend more time elaborating on their intervention itself, which will be much more useful to the reader. Why did it work so well? Were there institution-specific factors that played into its success? Are there ways in which it would, or wouldn’t, be easily adapted to other institutions or settings?
8. The discussion of Francini et al (citation 35) needs to be substantially revised. This was an observational study of BMA use and outcomes, and hence the association between BMA use and survival cannot be attributed to the drug. Randomized trials of BMAs have not demonstrated this survival benefit.
Minor comments:
1. Page 2, line 59: bone metastases can lead to SREs, but do not in and of themselves cause osteoporosis/osteopenia.
2. Authors should clarify and distinguish “inappropriate BMA use” from “inappropriate BMA dose.” Are these being using interchangeably (if so, it is confusing), or are these separate measures?
3. “we are seeing trends of significant improvement in the OS in patients both in the castration-sensitive and resistant settings.” This is confusing and borderline incorrect; it makes it sound like these are different groups of patients. No one dies of castration sensitive prostate cancer! It would be fine just to say something like “…significant improvement in OS for patients with advanced prostate cancer”
Author Response
File attached

Reviewer 2 Report
The authors should be mandated for having assessed this topic which is of main importance for patients under ADT. While dental evaluation and appropriate dosage of BMA is of interest, the main caveat in the clinical practice is the lack of DEXA scan measure before initiation of ADT (and during follow-up). Many practicians do not prescribe any DEXA scan, neither refer to any rheumatologist. This should have been emphasized, and the algorithm could have been adapted to improve this pathway. Yet, the authors focused on the lack of DEXA scan at follow-up which is also a point of interest.
Moreover, the risk of OJN is scare in PCa patients under BMA, and this risk could be more nuanced.
Methods : the authors should explain what they mean by “dental clearance” at line 127, page 4.
Results : line 138-139 indicate the percentage for the 29 and 12 remaining patients.
In the Figure, it is indicated that practicians were asked on the use of “leuprolide” in patients. May the authors explain why leuprolide (rather than another agonist or antagonist of LHRH) may pose a problem on bone density ?
Discussion : a chapter should be added the discuss the non-compliance to DEXA scan pathway before initiation of ADT, which is , from my point of view, the main limitation in taking care of PCa patients under ADT. This should appear more clearly.
Author Response
File attached

Round 2
Reviewer 1 Report
Overall the manuscript is much improved. The new description of the algorithm, visualized in Figure 2, vastly improves clarity for the reader in terms of what was done. Really my only remaining comment – which I have to get picky about – is the description of BMA-related benefits on page 9. Neither of these studies (Fancini and Saad) were randomized controlled trials of BMAs, and hence neither can support the conclusion of a causal impact of BMAs on survival; it would remain an equally viable hypothesis that physicians more frequently prescribe BMAs for patients that we expect to live longer, have better “substrate”, etc. But the authors use clearly causal language here – “Addition of BMA led to significantly longer OS.” For observational studies such as these, the correct interpretation of the results would be that “addition of BMA was associated with significantly longer OS.”
Author Response
Reviewer’s comments: Overall the manuscript is much improved. The new description of the algorithm, visualized in Figure 2, vastly improves clarity for the reader in terms of what was done. Really my only remaining comment – which I have to get picky about – is the description of BMA-related benefits on page 9. Neither of these studies (Fancini and Saad) were randomized controlled trials of BMAs, and hence neither can support the conclusion of a causal impact of BMAs on survival; it would remain an equally viable hypothesis that physicians more frequently prescribe BMAs for patients that we expect to live longer, have better “substrate”, etc. But the authors use clearly causal language here – “Addition of BMA led to significantly longer OS.” For observational studies such as these, the correct interpretation of the results would be that “addition of BMA was associated with significantly longer OS.”
Authors’ response: Thank you for the suggestion. We have changed the sentence as suggested.